# Evaluation of Changes in Gut Microbiota in Patients with Crohn’s Disease after Anti-Tnfα Treatment: Prospective Multicenter Observational Study

**DOI:** 10.3390/ijerph17145120

**Published:** 2020-07-15

**Authors:** Laura Sanchis-Artero, Juan Francisco Martínez-Blanch, Sergio Manresa-Vera, Ernesto Cortés-Castell, Josefa Rodriguez-Morales, Xavier Cortés-Rizo

**Affiliations:** 1Inflammatory Bowel Disease Unit. Department of Digestive Diseases, Hospital of Sagunto, Av. Ramón y Cajal s/n, 46520 Sagunto, Valencia, Spain; laurasanchisartero@gmail.com (L.S.-A.); prmorales06@yahoo.es (J.R.-M.); xacori@gmail.com (X.C.-R.); 2Genomics Laboratory. ADM-Lifesequencing. Parque Científico Universidad de Valencia. Catedrático Agustín Escardino Benlloch, 9. Edificio 2, 46980 Paterna, Valencia, Spain; sergiomanre@gmail.com; 3Department of Pharmacology, Pediatrics and Organic Chemistry Miguel Hernández University, Carretera de Valencia—Alicante S/N, 03550 San Juan de Alicante, Alicante, Spain; ernesto.cortes@umh.es

**Keywords:** gut microbiota, anti-TNFα, Crohn’s disease, *Faecalibacterium prausnitzii*, *Escherichia coli* and *Clostridium coccoides* group

## Abstract

*Background:* Crohn’s disease is believed to result from the interaction between genetic susceptibility, environmental factors and gut microbiota, leading to an aberrant immune response. The objectives of this study are to evaluate the qualitative and quantitative changes in the microbiota of patients with Crohn’s disease after six months of anti-tumor-necrosis factor (anti-TNFα) (infliximab or adalimumab) treatment and to determine whether these changes lead to the recovery of normal microbiota when compared to a control group of healthy subjects. In addition, we will evaluate the potential role of the *Faecalibacterium prausnitzii/Escherichia coli* and *Faecalibacterium prausnitzii/Clostridium coccoides* ratios as indicators of therapeutic response to anti-TNFα drugs. *Methods/Design:* This prospective multicenter observational study will comprise a total of 88 subjects: 44 patients with Crohn’s disease scheduled to start anti-TNFα treatment as described in the drug specifications to control the disease and 44 healthy individuals who share the same lifestyle and eating habits. The presence of inflammatory activity will be determined by the Harvey-Bradshaw index, analytical parameters in blood, including C-reactive protein, and fecal calprotectin levels at commencement of the study, at three months and at six months, allowing the classification of patients into responders and non-responders. Microbiota composition and the quantitative relationship between *Faecalibacterium prausnitzii* and *Escherichia coli* and between *Faecalibacterium prausnitzii* and *Clostridium coccoides*
*group* as indicators of dysbiosis will be studied at inclusion and six months after initiation of treatment using ultra sequencing with Illumina technology and comparative bioinformatics analysis for the former relationship, and digital droplet PCR using stool samples for the latter. Upon inclusion, patients will complete a survey of dietary intake for the three days prior to stool collection, which will be repeated six months later in a second collection to minimize dietary bias. *Discussion:* In this study, massive sequencing, a reliable new tool, will be applied to identify early biomarkers of response to anti-TNF treatment in patients with Crohn’s disease to improve clinical management of these patients, reduce morbidity rates and improve efficiency.

## 1. Background

Gut microbiota is defined as the total number of living microorganisms (bacteria, fungi, archaea, viruses and others) present in the intestine. The gut microbiome is a taxonomic characterization of microbial diversity including the set of genomes and, via genes, their physicochemical capabilities [1,2]. In healthy individuals, the gut bacterial microbiota is composed of more than 10^18^ different microorganisms. The vast majority of these are bacteria, with some 1100 more prevalent species; of these an estimated 160 species of bacteria are specific to each individual [3].

The microbiome, therefore, is a highly complex structure, involving thousands of microorganisms belonging to very different taxonomic classifications and, consequently, millions of associations between them, making its study a great challenge [4]. Through advances in bioinformatics, in 2003 the human genome was decoded, a milestone in science. Since then, much attention has been focused on deciphering this extensive network of microbes, the microbiota, also known as the second human genome, or as another organ. These microbes coexist with us and have a larger total number of genes than the human genome. In short, there is no simple description of these structures. However, due to their importance, considerable interest has been generated in the identification of patterns associated with human health and disease states which may even lead to the development of microbiota-based diagnostics and therapies as well as having implications for nutritional or pharmaceutical interventions [2]. To this end, reproducible patterns of gut microbiome variation have been observed in healthy adults, determining the existence of three major microbiota communities, based on the predominance or absence of species of the key genera [5]. Any other combination of key genera or genera not described by Arumugam et al. [5], together with a reduction in biodiversity, is considered dysbiosis [6,7]. Numerous technologies have been applied with the aim of examining the gut microbiota, which has resulted in the capacity and cost of microbiota research being significantly reduced in recent years, mainly due to advances in massive sequencing technologies such as next-generation sequencing. These techniques allow information of interest to be obtained quickly and efficiently by sequencing regions of the prokaryotic 16S ribosomal RNA gene [8].

In inflammatory bowel disease (IBD), the role of the microbiota in disease development and onset is very clear. While it is true that to date no specific pathogen has been conclusively identified as a trigger, we do know that, for the disease to develop, dysbiosis or a definitive change in the intestinal microbiota must occur and is likely to be the defining event in the development of Crohn’s disease (CD). In IBD, it has been documented that the gut microbiota bacterial composition transitions from saprophytic to predominantly pathogenic [9]. Indeed, there is evidence of a significant increase in *Escherichia coli* concentrations, including pathogenic variants, in CD patients with ileal involvement [10]. It unknown at present whether dysbiosis is a cause or a consequence of the development of CD. It appears that the combination of a genetic predisposition and an alteration in gut microbiota are the final triggers of a chronic IBD-type inflammatory process. Specifically, we know that the disease develops in genetically susceptible individuals through dysregulation of homeostasis between commensal microbiota and/or other environmental elements and an altered immune response in the patient. An error in the interpretation of the stimulus or in the regulation of the immune response leads to an imbalance between pro- and anti-inflammatory factors, perpetuating the inflammatory process [11].

The gut microbiota play a crucial role in the development of the immune system and maintenance of the intestinal epithelial barrier. Inflamed ileal mucosa in CD patients shows increased production of tumor necrosis factor-α (TNFα), compared to normal ileum, induced by a dysbiosis in the gut microbiota, with a significant increase in bacteria that stimulate TNF production. Numerous bacteria in the commensal microbiota inhibit the release of TNF and other pro-inflammatory cytokines, e.g., bacteria that produce short-chain fatty acids (SCFA), inducing a potent anti-inflammatory effect in the intestinal mucosa. In contrast, other types of bacteria, such as *Escherichia coli ECOR-26*, which have been linked to CD, induce increased TNF release and stimulation of IL-10 release [12]. Current hypotheses in favor of a higher release of TNFα induced by intestinal dysbiosis support the idea that restoration of a less pathogenic microbe in the intestinal mucosa (inducing a reduced release of TNF and other pro-inflammatory cytokines) could lead to better disease control [13].

Today, anti-TNF therapy is one of the therapeutic pillars in the management of CD, but these drugs only treat the consequences of the disease and not the possible cause. Approximately one quarter of CD patients will be primary non-responders to anti-TNF agents, and one third of responders will experience a loss of response over time [14]. To improve treatment effectiveness, it is essential to study why these patients do not have an optimal response.

Traditionally, the prognosis and monitoring of treatments in patients with CD have been limited to the control of clinical symptoms (e.g., through the Harvey-Bradshaw Index [HBI]: see Table 1) accompanied by imaging techniques (primarily endoscopy and magnetic resonance imaging). However, these tools for assessing disease activity have many drawbacks and limitations. Among them, clinical scoring systems are highly subjective and can be misleading in this disease, characterized by alternating periods of exacerbation and remission [15,16]. Ileo-colonoscopy with biopsies is the current gold standard for the diagnosis and evaluation of inflammatory activity, with the great disadvantage that it is an invasive procedure [17] and that it is not always possible to reach the diseased area. In an attempt to identify non-invasive markers, fecal calprotectin (FC), a protein originating from the migration of neutrophils to the intestinal mucosa, was introduced as an indirect trait of intestinal inflammation, allowing more objective monitoring than clinical indices, although with low specificity and low positive predictive value depending on the chosen cut-off point [18,19].

Currently, the role of bacterial gut microbiota is described as a key factor in the development of CD. Various authors defend the reduction in biodiversity and abundance of the phyla *Bacteroidet*es and *Firmicutes* such as *Faecalibacterium prausnitzii* (SCFA-producing bacteria), as well as an increase in the phylum *Proteobacteria* such as *Escherichia coli*, characteristic of patients with this disease compared to healthy individuals [6,7,20]; a decrease in abundance of this species has even been observed after anti-TNF treatment [20]. In this line, one study has identified certain specific microbial profiles that correlate with the recurrence of disease after achieving remission with infliximab treatment [14]. Several studies have also shown that the greater abundance of SCFA-producing bacteria predicted the effectiveness of infliximab [21,22] and another study associated the greater abundance of SCFA with a sustained response to this treatment [23].

A greater understanding of the composition of the bacterial gut microbiota in CD patients, such as the persistence of a significant proportion of certain pathogenic bacteria or low bacterial biodiversity, would make it possible to determine the role of gut microbiota in therapeutic responses and to establish biomarkers of response and relapse, as well as to determine whether it is necessary to restore intestinal normo-biosis in these patients. In addition, different gut microbiota profiles can be found, which enable us to predict the response to different therapeutic lines, thus being more efficient from the outset. Accordingly, this study has been designed to analyze biomarkers of response to anti-TNF treatment in CD through gut microbiota as an alternative non-invasive tool for predicting treatment response.

## 2. Methods

### 2.1. Ethics, Consent and Permission

The final protocol was approved by the Sagunto Hospital Clinical Research Ethics Committee, in accordance with applicable national and local laws and requirements. The study was classified by the Spanish Agency for Medicine and Health Products as a prospective follow-up post-authorization study.

The study adheres to the European guidelines for the protection of human research subjects, the Declaration of Helsinki and the recommendations of the European Network of Centres for Pharmacoepidemiology and Pharmacovigilance. Approval was obtained from all local ethics committees at all participating centers. Prior to inclusion, each patient will receive a detailed report on the nature, scope and possible consequences of the study from a physician and then will provide written informed consent. No action specifically required for the study will be taken without the valid consent of the patient.

### 2.2. Investigators

This multicenter study in the Valencian Community (Spain) includes eight academic medical centers. The patients will be recruited by the respective participating centers of the Public Health System of the Valencian Community, in which there are approximately 15,000 patients with IBD [24]. All the researchers will be gastroenterologists with experience in the follow-up and treatment of patients with CD. The centers participating in the study will have a total load of 3600 patients with CD, 10–15% of whom will be candidates for biological treatment according to estimated data from each center.

### 2.3. Study Objectives

The main objective of this study is to evaluate the modification of the gut microbiota in CD patients prior to and six months after anti-TNF therapy (infliximab or adalimumab). The secondary objectives are to evaluate the association between the changes in microbiota and the clinical, biological and endoscopic response of the patients; to correlate the normalization of the gut microbiota with the response to anti-TNF treatment; to determine the level of biodiversity of the fecal microbiota at the inclusion and completion of the study in each participant; to evaluate the potential role of the *Faecalibacterium prausnitzii/Escherichia coli* and *Faecalibacterium prausnitzii/Clostridium coccoides* ratios as an indicator of therapeutic response; and finally, to describe the clinical, biological and epidemiological characteristics of the patients included in the study prior to and six months after anti-TNF treatment.

### 2.4. Primary Study Variable

Normalization of gut microbiota after treatment: percentage of patients with dysbiosis of the gut microbiota before the introduction of anti-TNF therapy whose microbiota is normalized after six months of treatment. Dysbiosis is defined as a gut microbiota pattern different from the established patterns of normality according to Arumugam et al. [5]: enterotype 1 (ET B) predominantly contains *Bacteroides*, enterotype 2 (ET P) is characterized by the high abundance of *Prevotella* inversely correlated with *Bacteroides*, and enterotype 3 (ET F) can be distinguished by the presence of *Firmicutes*, highlighting the genus *Ruminococcus*.

### 2.5. Secondary Study Variables

Percentage of patients with dysbiosis at inclusion (after initiation of anti-TNF treatment) and at study completion, measured as dichotomous qualitative variables.

Levels of biodiversity of the fecal microbiota for the microbiome analyzed are defined as follows: low biodiversity is a total of five or fewer species, medium biodiversity is a total of between 6 and 10 species, and high biodiversity is a total of 11 or more species present.

Determine the increase in biodiversity pre- and post-anti-TNF treatment, defining an improvement as an increase in the number of species greater than or equal to five with respect to baseline.

Determine the relationship between *Faecalibacterium prausnitzii*/*Escherichia coli* and *Faecalibacterium prausnitzii*/*Clostridium coccoides* pre- and post-anti-TNF treatment.

Associate the presence and type of modifications in the gut microbiota after anti-TNF treatment with the clinical and biological response of the patient during the study.

Definitions:○Clinical remission: HBI ≤4 (Table 1)○Clinically active disease: HBI >4○Clinical response: when the HBI falls by three or more points.○Relapse: increased activity assessed by clinical, laboratory, radiology or endoscopic findings leading to a change in treatment to control the disease or an HBI > 4.○Biological remission: C-reactive protein (CRP) < 5 mg/L and (FC) < 250 μg/g.○Active biological disease: CRP ≥ 5 mg/L and FC ≥ 250 μg/g.○Overall response: the evaluation of clinical response will be established subjectively by the responsible physician according to clinical and analytical parameters, classifying patients as non-responders, responders without remission and responders in remission.

Clinical, demographic and complementary test variables included in the study protocol (Table 2):○Epidemiological characteristics: age, sex, smoking habits and body mass index.○Clinical characteristics: date of diagnosis and disease pattern, Montreal classification, activity index, presence of extra-intestinal manifestations, pharmacological history, concomitant medication, adverse effects, presence/absence of initial and final overall response, and clinical decisions derived from this response.○Anti-TNF treatment data: indication, type, start date, induction pattern, maintenance pattern, specify whether anti-TNF drugs were used prior to 24 weeks before inclusion and reason for discontinuation.○Complementary examinations for each patient:-Laboratory tests: complete blood count, erythrocyte sedimentation rate, CRP, fibrinogen, ferritin, transferrin saturation index, total proteins, albumin, urea, creatinine, GOT/AST (Aspartate Aminotransferase), GPT/ALT (Alanine Aminotransferase), GGT (Gamma-glutamyl transpeptidase), ALP (Alkaline Phosphatase), cholesterol and triglycerides-FC-Stool culture, parasites and *Clostridium difficile* toxin in feces-Stool collection for microbiota analysis-Radiological (ultrasound/computerized tomography/magnetic resonance imaging) and/or endoscopic testing if available prior to treatment (at least 12 weeks prior to inclusion) or 6 months after treatment.-72-h dietary record prior to stool sample collection for microbiota analysis. Patients will be provided with a daily survey in which they must record food and beverage intake, specifying characteristics, quantity and brands of packaged products.-Record of adherence to the Mediterranean diet. Together with the dietary record, the patients will also complete a validated survey of adherence to the Mediterranean diet [25] classifying this adherence as low (0–6 points), moderate (7–10 points) or high (11–14 points).

### 2.6. Study Design

This prospective, observational, multicenter study will include patients with CD who require anti-TNF therapy to control IBD. Anti-TNF (infliximab or adalimumab) treatment will be initiated under medical prescription according to the drug data sheet and the MAISE (drugs with a high health or economic impact) of the Valencian Health Agency. Administration of treatment will be performed in routine clinical practice and will not be promoted by this study. Fecal samples will be collected for gut microbiota analysis prior to exposure to the anti-TNF drug and six months after commencement of treatment. The patients will be monitored prospectively by their responsible physician during routine clinical practice during the first six months of anti-TNF treatment, recording the presence of inflammatory activity and whether there was no response, partial response or clinical remission at three months and at the end of the study follow-up.

Data to be recorded will include the presence of inflammatory activity (through calculation of the HBI), laboratory analytical data (complete blood count, general biochemistry, CRP) and the FC value prior to and three and six months after anti-TNF drug exposure, as well as treatments at the time of inclusion and any treatment changes throughout the duration of the study. The patient will be instructed to avoid taking probiotics and antibiotics during the study period. However, if antibiotics have to be taken, the patient must contact the study coordinator. In all cases, the second stool sample for mass sequencing will be performed after four weeks without treatment with probiotics/antibiotics, maintaining the same dietary conditions to avoid variability.

Whenever available, radiological and endoscopic activity data used in clinical practice to assess disease activity will be collected when performed, up to 12 weeks prior to patient inclusion and during the study.

Patients who undergo surgical resection within the study period will remain included but will be classified as non-responders to anti-TNF. Prior to surgery, clinical evaluation will be performed and blood and stool samples collected for fecal microbiota analysis, identical to the last determination at week 24 performed per protocol.

The patients will be provided with a 72-h dietary survey designed to record their dietary intake three days prior to stool collection at inclusion, repeated at six months to minimize bias concerning diet. Similarly, patients must complete a validated survey of adherence to the Mediterranean diet at inclusion and at six months to assess significant changes in their eating habits. An outline of the different data collection phases is shown in Figure 1.

### 2.7. Inclusion Criteria

All the patients must be adults (aged ≥ 18 years) diagnosed with CD with a clinical course of moderate-severe disease requiring anti-TNF treatment, in whom the indication for such treatment was inflammatory bowel activity.

### 2.8. Exclusion Criteria

Excluded from the study will be all patients who had taken antibiotics, probiotics, or proton pump inhibitors four weeks prior to the start of the study and stool collection; patients with chronic hepatitis C virus and chronic HIV infection; indication for anti-TNF treatment for reasons other than control of their luminal disease (e.g., enteropathic arthropathy, perianal disease, or prevention of recurrence); patients with previous ileum or colon surgery; and patients who will have received previous anti-TNF treatment in the 24 weeks prior to the start of the study.

### 2.9. Sample Size Considerations

Few studies have evaluated the variability in gut microbiota after anti-TNF therapy in adult patients with IBD [14,21,22,23,26,27,28,29,30]. However, from the results obtained it is not possible to determine the exact percentage of expected variability. Accepting an alpha risk of 0.05 and a beta risk of 0.2 in a bilateral contrast, 44 subjects will be required assuming that the initial proportion of events is 0.95 (degree of dysbiosis in the population at the start of the study) and 0.70 at the end. A loss to follow-up rate of 10% has been estimated.

### 2.10. Planning of the Sample Collection

The participating centers will be responsible for taking samples and determining conventional analytical and fecal data (complete blood count, stool samples, parasitology, FC, etc.) within routine clinical practice. The specific samples for the analysis of microbiota to be conducted externally will be collected by the patients themselves. The samples must be frozen after collection (by the patient in his or her freezer) until they are delivered to the laboratory of the hospital of origin, where they will be stored at −20 °C. The samples will subsequently be centralized.

### 2.11. Sequencing and Bioinformatics

The samples will be coded and the bacterial microbiota present will be analyzed using capture of the v3-v4 region of the 16S rRNA subunit [8], ultra-sequencing with Library Illumina 15044223 B protocol (ILLUMINA) comparative bioinformatics analysis. From 200 mg of stool, DNA will be obtained through a combination of mechanical and enzymatic lysis, and purified using the PowerFecal Pro DNA isolation (Qiagen) protocol with modifications. The DNA will be processed and prepared for sequencing. The sequences obtained will be filtered by parameters of quality (threshold value Q20) and length (sequences greater than 250 nucleotides). This strategy avoids erroneous ascriptions that generate an incorrect distribution of taxa. The minimum number of readings per sample will be 5000 and the mean length greater than 400 nt. A rarefaction analysis will be performed on the sample sequences to assess saturation for microbial biodiversity. Subsequently, the analysis of taxonomic identification at different taxonomic levels will be performed using the Microbiome bioinformatics with QIIME 2 2019.4 protocol [31] (Figure 2). 

### 2.12. Identification and Evaluation of Potential Biomarkers

To identify potential non-invasive biomarkers from the characterization of the intestinal microbiome in various stages of CD, we will use LEfSe (Linear Discriminant Analysis Effect Size), an algorithm designed for the discovery of metagenomic biomarkers through class comparison, biological consistency testing and effect size estimation [32].

Due to the need to develop a rapid method of analysis such as the ratios of *Faecalibacterium prausnitzii/Escherichia coli* and *Faecalibacterium prausnitzii/Clostridium coccoides* group, a triplex digital droplet PCR will be implemented to allow the absolute quantification of the *Faecalibacterium prausnitzii*, *Escherichia coli* and *Clostridium coccoides* species, as well as the total number of bacteria present in a given sample through the 16S rRNA gene, all with a minimum volume and in a single reaction. Therefore, the appropriate primers and probes have been selected to perform this digital droplet PCR (Table 3), as well as to optimize the different concentrations and hybridization temperatures, based on numerous relevant studies [20,33,34,35,36].

### 2.13. Anti-TNF Dosage and Safety Evaluation

Anti-TNFs are monoclonal antibodies that neutralize pro-inflammatory cytokine tumor necrosis factor (TNF) involved in the inflammatory cascade of CD and other immune-mediated diseases. The anti-TNFs used are infliximab and adalimumab by indication for induction of remission in patients with moderate-severe CD. The administered doses of anti-TNFs are those listed in the data sheet, both at induction and maintenance doses. Infliximab dose regimen for adult patients with CD 5 mg/kg is given as an IV induction regimen at 0, 2, and 6 weeks followed by a maintenance regimen of 5 mg/kg IV every 8 weeks thereafter; treatment with 10 mg/kg IV may be considered for patients who respond and then lose their response. Adalimumab dose regimen for adult patients with CD is 160 mg initially on Day 1 (given in one day or split over two consecutive days), followed by 80 mg two weeks later (Day 15). Two weeks later (Day 29) they begin a maintenance dose of 40 mg every other week. During the prospective follow-up, according to usual clinical practice, the gastroenterologist responsible will modify or maintain a maintenance pattern according to the response to the drug by means of clinical (HBI) and analytical (fecal C-reactive protein and calprotectin) variables, which will be recorded in each patient’s data collection notebook.

To identify any adverse effects associated with the administration of the anti-TNFα drug, a combination of physical examination, recording of vital signs (BP, pulse, temperature), and questionnaires are evaluated after administration (30 min later) and at follow-up visits at weeks 12 and 24. These questionnaires inquire about possible adverse effects related to adverse reactions to infliximab and adalimumab drugs (nausea, headache, dizziness, fever, hives, reactions at the infusion site, nervous system disorders, cardiac arrhythmia or myocardial ischemia, hypertensive or hypotensive events, skin reactions, gastrointestinal disorders, infections, hypersensitivity or anaphylaxis). In addition, patients are asked open-ended questions about their general well-being to request notification of any other adverse effects not listed in the data sheet. In order to assess the causality of adverse effects related to infliximab and adalimumab, the ADR (Adverse Drug Reaction) Naranjo probability scale is applied [37].

Laboratory parameters, including white blood cell count, liver transaminase levels, phosphate levels, and kidney function are studied prior to anti-TNF treatment and at the 12-week and 24-week follow-up (post-anti-TNF).

### 2.14. Statistical Analysis

As this is an observational non-interventional study, there may be confounding factors between treatment allocation and outcomes in analyses of comparative response rates and samples may be included of patients that do not represent real-world clinical practices. Accordingly, all analyses in this study will be considered to be of an exploratory nature. Statistical analysis will be performed with SPSS v.22 software version 9.2 or later.

Population characteristics (including demographic characteristics, medical conditions, disease duration, types of treatment used at the start of the study and other variables collected in the data collection logbook) and all primary and secondary endpoints will be summarized by indicating the mean, standard deviation, minimum value, maximum value, median, 25th and 75th percentile and 95% confidence interval (CI) of the mean for continuous variables, and the absolute and relative frequencies, with a 95% CI of the proportion for categorical data. In the bivariate analysis, the chi-square test will be used to determine differences in proportions and Student’s *t*-test for paired data before and after the administration of anti-TNF treatment.

The statistical analyses conducted throughout this study on bioinformatics data management to determine significant differences between established groups, their corresponding graphs and the study of ROC curves (AUC values) will be performed using GraphPad Prism software version 8.2.0. STAMP (Statistical Analysis of Metagenomic Profiles), another bioinformatics tool based on Python, will also be used. STAMP was specifically designed for statistical processing and creation of plots from large amounts of biological data [38].

## 3. Discussion

The gut microbiome has a highly complex structure, involving thousands of microorganisms belonging to very different taxonomic classifications with millions of relationships between them, making its study a great challenge. In IBD, no specific pathogen has been definitively identified, but the gut microbiota as a whole has been shown to be pathogenic, contributing to the development of a deregulated inflammatory response in susceptible hosts. Several authors defend the suggestion that there is a reduction in biodiversity and abundance of *Bacteroidetes* and *Firmicutes*, as well as an increase in *Proteobacteria*, characteristic of patients compared to healthy individuals [6,7].

Treatment for CD is selected according to the severity of the disease and the response to previous therapies. Among the drugs used are biological treatments based on TNFα monoclonal antibodies such as infliximab and adalimumab, developed for the induction and maintenance of remission, allowing the control of symptoms and an improvement in the quality of life of responders, as well as changes in the natural course of the disease [38,39,40]. Nonetheless, although approximately one third of patients do not respond to these inhibitory therapies, we currently do not have a non-invasive biomarker that serves as a tool to predict this response, and invasive methods such as colonoscopy continue to be the gold standard for assessing therapeutic response.

In this prospective observational study, a new tool, massive sequencing of bacterial DNA, will be applied to the study of a clinical problem that affects an important number of patients. This will enable both the identification of early biomarkers of response in patients with CD after anti-TNF treatment as well as the prediction of therapeutic response from the start to thus improve the clinical management of these patients, reduce morbidity rates and increase efficiency. Since this is a longitudinal study, the patients will be analyzed before and after exposure to anti-TNF treatment, and the data will be paired, thereby diminishing the effect of the high variability of gut microbiota.

We know that the composition of the microbiota varies due to multiple external factors, particularly diet, and that a dietary intervention of just three days can cause a change in enterotype [41,42]. Nevertheless, after ten days, the enterotypes stabilized in one study suggesting a tendency to return to the original state [41]. Even so, we added to the protocol a 3-day dietary record prior to stool collection for microbiota analysis and will repeat this dietary record prior to the second assessment after six months of anti-TNF treatment to minimize significant diet-induced changes. An additional measure to objectively determine whether significant changes in the eating behavior of patients occur during the six months of the study will be undertaken through completion of a validated survey on adherence to the Mediterranean diet, both at inclusion and completion of the study. Given that long-term disturbances have a more profound effect, with a one-year diet modification having a strong impact on the *Bacteroidetes/Firmicutes ratio* [41,43], this may lead to changes in enterotype.

The proposed design takes diet into consideration and is therefore novel with respect to similar studies published to date, which did not evaluate this factor known to modify the composition of the microbiota. This study may provide additional evidence regarding potential non-invasive tools such as biomarkers of the response before and after anti-TNF therapy in CD as a starting point for future clinical trials. These trials could determine the most effective treatment among not only these therapies but all therapies used in the management of CD based on patient microbiota and provide more appropriate, inexpensive and non-invasive tools for predicting clinical response to treatment.

## 4. Conclusions

Currently, the role of bacterial gut microbiota is described as a key factor in the development of Crohn’s disease, we do know that, for the disease to develop, dysbiosis or a definitive change in the intestinal microbiota must occur. This study may provide additional evidence regarding potential non-invasive tools such as microbiota-based biomarkers of the response before and after anti-TNF therapy in Crohn’s disease as a starting point for future clinical trials. These trials could determine the most effective treatment among not only these therapies but all therapies used in the management of Crohn’s disease based on patient microbiota and provide more appropriate, inexpensive and non-invasive tools for predicting clinical response to treatment, reduce morbidity rates and improve efficiency.

## Figures and Tables

**Figure 1 ijerph-17-05120-f001:**
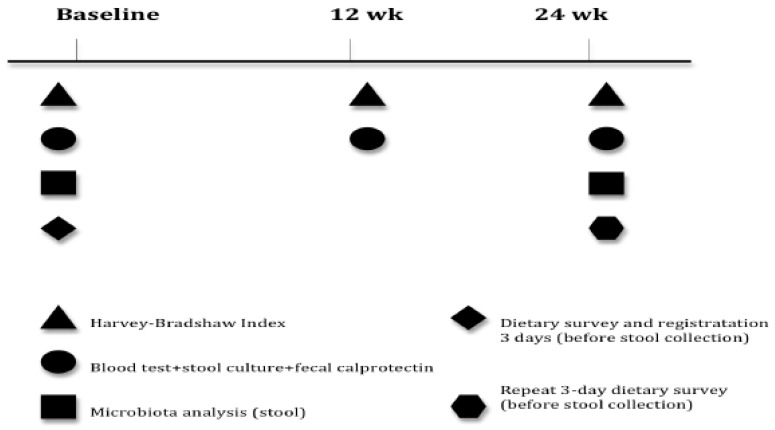
Overall study design.

**Figure 2 ijerph-17-05120-f002:**
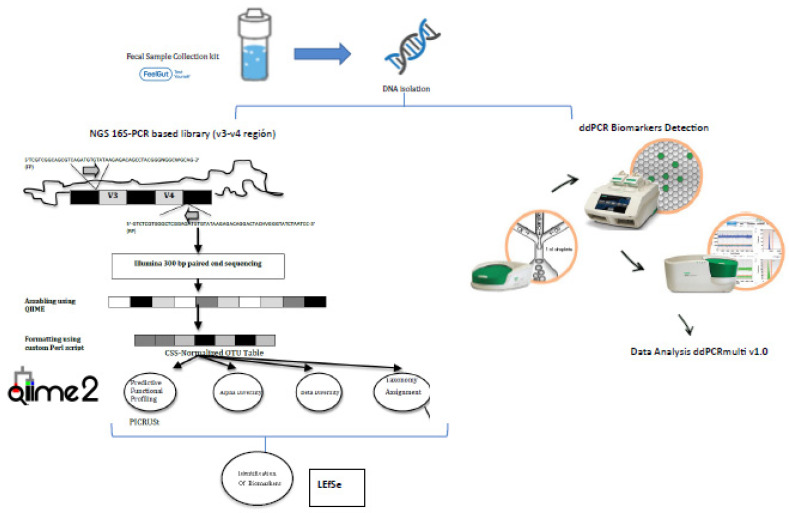
Sequencing and bioinformatics scheme.

**Table 1 ijerph-17-05120-t001:** Harvey-Bradshaw Index (HBI).

**General Well-Being (Previous Day)**	0 (Very Well)	1 (Slightly Below Par)	2 (Poor)	3 (Very Poor)	4 (Terrible)
**Abdominal pain**	0 (none)	1 (mild)	2 (moderate)	3 (severe)	
**Number of liquid or soft stools per day (previous day)**					
**Complications** **(score 1 per item)**	Arthritis/ arthralgia	Iritis/ uveitis	Erythema nodosum/ aphthous ulcers/Pyoderma gangrenosum	Anal fissure new fistula/ abscess	
**Abdominal mass**	0 (none)	1 (dubious)	2 (definite)	3 (definite + tender)	

Remission ≤ 4 points; mild disease 5–6 points; moderate disease 6–12 points; severe disease > 12 points.

**Table 2 ijerph-17-05120-t002:** Summary of variables to be collected during the study.

Study Variables
**Primary:** Normalization of gut microbiota (after anti-TNF treatment) (yes/no)
**Secondary variables:** **A. Gut microbiota** - Initial dysbiosis (yes/no) - Final dysbiosis (yes/no) - Initial biodiversity level (low/medium/high) - Final biodiversity level (low/medium/high) - Increase in biodiversity (yes/no) - Initial *Faecalibacterium prausnitzii/Escherichia coli* - Final *Faecalibacterium prausnitzii/Escherichia coli* - Initial *Faecalibacterium prausnitzii/Clostridium coccoides* group - Final *Faecalibacterium prausnitzii/Clostridium coccoides* group **B. Clinical-biological parameters:** - Initial Harvey-Bradshaw Index - Final Harvey-Bradshaw Index - Initial C-reactive protein (mg/l) - Final C-reactive protein (mg/l) - Initial fecal calprotectin (µg/g) - Final fecal calprotectin (µg/g) **C. Epidemiological data** - Age (years) - Sex (male/female) - Time from diagnosis to anti-TNF treatment (months) - Body Mass Index (kg/m^2^) - Smoking habit (yes/no) - 72-h dietary record - Adherence to Mediterranean diet

**Table 3 ijerph-17-05120-t003:** Primers and probes used in digital droplet PCR [20,35,36].

Target	Primers and Probe	Sequences 5′-3′	Reference
*F. prausnitzii*	Fpra_428_F	TGTAAACTCCTGTTGTTGAGGAAGATAA	[20]
	Fpra_583_R	GCGCTCCCTTTACACCCA	
	Fpra_493_PR	FAM/CAAGGAAGTGACGGCTAACTACGTGCCAG/IABkFQ	
*E. coli*	Ecoli_395_F	CATGCCGCGTGTATGAAGAA	[35]
	Ecoli_490_R	CGGGTAACGTCAATGAGCAAA	
	Ecoli_437_PR	FAM/TATTAACTTTACTCCCTTCCTCCCCGCTGAA/IABkFQ	
*C. coccoides*	F_Ccoc_07	GACGCCGCGTGAAGGA	[36]
	R_Ccoc_14	AGCCCCAGCCTTTCACAT	
	P_Erec_482	VIC/CGGTACCTGACTAAGAAG/IABkFQ	
*Bacteria*	F_Bact_1369	CGGTGAATACGTTCCCGG	[36]
	R_Prok_1492	TACGGCTACCTTGTTACGACTT	
	P_TM_1389F	FAM/CTTGTACACACCGCCCGTC/IABkFQ	

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
