# Peer review of "Evaluation of Changes in Gut Microbiota in Patients with Crohn’s Disease after Anti-Tnfα Treatment: Prospective Multicenter Observational Study"

_ijerph, 2020, doi:10.3390/ijerph17145120_

Round 1
Reviewer 1 Report
The authors present a study that evaluates changes in gut microbiota in patients with Crohn’s Disease after anti-TNF Alpha treatment. The content of the study is of high clinical interest and involves a hot topic in the management of CD. The prospective observational multi-centric study design is an excellent choice.
Nevertheless I have 2 questions/remarks:
1) What happens with patients that undergo surgical resection within the study period?
2) What happens with patients that take antibiotics within the study period? Is probiotic intake registered?
Author Response
1) What happens with patients that undergo surgical resection within the study period?
In response to your comment, we added this information in the Study design section:
“Patients who undergo surgical resection within the study period will remain included but will be classified as non-responders to anti-TNF. Prior to surgery, clinical evaluation will be performed and blood and stool samples will be collected for fecal microbiota analysis, identical to the last determination at week 24 performed per protocol”.
2) What happens with patients that take antibiotics within the study period? Is probiotic intake registered?
In response to your comment, we added this information in the Study design section:
“The patient will be instructed to avoid taking probiotics and antibiotics during the study period. However, if antibiotics must be taken, the patient should contact the study coordinator. In all cases, the second stool sample for mass sequencing is performed after 4 weeks without treatment with probiotics/antibiotics and maintaining the same dietary conditions to avoid variability”.

Reviewer 2 Report
The present project report described a human trial in which evaluated the changes in gut microbiota in patients with crohn’s disease after anti-TNFα treatment. It looks like this report is close to a research proposal, but not a comprehensive report. As I didn’t find any research outcome in this report. In addition, the most key details about the usage of anti-TNFα agents is missing. How to evaluate the toxicity, allergy, and the safety of anti-TNFα? How to determinate the dosages? Which kind of anti-TNFα agents were used, small molecules, or antibodies? Have you test the bioavailability of these regents? In addition, I didn’t find enough pre-clinical data about the usage of anti-TNFα treatment against IBD.
Author Response
- How to evaluate the toxicity, allergy, and the safety of anti-TNFα?
According to your recommendations, we added this information in the Study design section:
“To identify any adverse effects associated with the administration of the anti-TNFα drug, a combination of physical examination, recording of vital signs (BP, pulse, temperature), and questionnaires are evaluated after administration (30 minutes later) and at follow-up visits at weeks 12 and 24. These questionnaires inquire about possible adverse effects related to adverse reactions to infliximab and adalimumab drugs (nausea, headache, dizziness, fever, hives, reactions at the infusion site, nervous system disorders, cardiac arrhythmia or myocardial ischemia, hypertensive or hypotensive events, skin reactions, gastrointestinal disorders, infections, hypersensitivity or anaphylaxis). In addition, patients are asked open-ended questions about their general well-being to request notification of any other adverse effects not listed in the data sheet. In order to assess the causality of adverse effects related to infliximab and adalimumab, the ADR Naranjo probability scale is applied [37].
Laboratory parameters, including white blood cell count, liver transaminase levels, phosphate levels, and kidney function are studied prior to anti-TNF treatment and at the 12-week and 24-week follow-up (post-anti-TNF)”.
-How to determinate the dosages?
“Anti-TNFs are monoclonal antibodies that neutralize pro-inflammatory cytokine tumor necrosis factor (TNF) involved in the inflammatory cascade of CD and other immune-mediated diseases. The anti-TNFs used are infliximab and adalimumab by indication for induction of remission in patients with moderate-severe CD. The administered doses of anti-TNFs are those listed in the data sheet, both at induction and maintenance doses. Infliximab dose regimen for adult patients with CD 5 mg/kg given as an IV induction regimen at 0, 2, and 6 weeks followed by a maintenance regimen of 5 mg/kg IV every 8 weeks thereafter; treatment with 10 mg/kg IV may be considered for patients who respond and then lose their response. Adalimumab dose regimen for adult patients with CD is 160 mg initially on Day 1 (given in one day or split over two consecutive days), followed by 80 mg two weeks later (Day 15). Two weeks later (Day 29) begin a maintenance dose of 40 mg every other week. During the prospective follow-up, according to usual clinical practice, the gastroenterologist responsible will modify or maintain a maintenance pattern according to the response to the drug by means of clinical (Harvey-Bradshaw index) and analytical (fecal C-reactive protein and calprotectin) variables, which will be recorded in each patient’s data collection notebook”.
- Which kind of anti-TNFα agents were used, small molecules, or antibodies?
Adalimumab and Infliximab
- Have you test the bioavailability of these regents? In addition, I didn’t find enough pre-clinical data about the usage of anti-TNFα treatment against IBD.
Our study, as mentioned above, follows a dosage pattern according to the technical data sheet within usual clinical practice; therefore, these pre-clinical data are included in the technical data sheet in the pharmacokinetic and pharmacodynamic profile of infliximab and adalimumab. If you consider it necessary, these data could be attached as an appendix.
The study does not contemplate the determination of serum levels, although in future studies, we believe it could be interesting to determine these levels to verify their association with the clinical efficacy of anti-TNF and its effect on intestinal microbiota.

Reviewer 3 Report
The manuscript ID ijerph-842229 titled “Evaluation Of Changes In Gut Microbiota In Patients With Crohn’s Disease After Anti-Tnfα Treatment: Prospective Multicenter Observational Study” by Sanchis-Artero and co-workers described a project report with 88 subjects, 44 of them with Crohn’s disease, and also reported the evaluation of blood parameters and microbiota. I have a few concerns about the present manuscript.
-What is the new in the study, Crohn’s disease microbiota is reported in several manuscripts?
-The strain names are not properly used, please italicize
-The reference format is not appropriate according to the journal guidelines
-The project was approved in 2015, in 2020 is finish or they are recruiting patients?
-Please define in material and methods, the Harvey-Bradshaw index
-What is the dietary record, 24h, or FFQ?
Author Response
-, Crohn’s disease microbiota is reported in several manuscripts?
Currently, there are numerous diseases where there seems to be a clear association between alterations in the intestinal microbiota and the development of these diseases (Alzheimer's, autism, depression, metabolic syndrome, ...) or the response to drugs (chemotherapy in colon cancer, melanoma, ...). Therefore, it is a field of research that in recent years and through progress in bioinformatics, is revolutionizing our understanding of many diseases. However, the great complexity of this type of study, which aims to analyze the intestinal microbiota and its relationship with the therapeutic response, often prevents us from reaching conclusions. We wanted to carry out a study design that can be applied by other researchers to other diseases. The model can be extrapolated to other lines of research related to the microbiota and clinical pathologies of other specialties. We ourselves had great concerns about how to implement this project and the design has allowed us to achieve good preliminary results.
-The strain names are not properly used, please italicize
The strain names have been italicized in the manuscript.
-The reference format is not appropriate according to the journal guidelines
The reference format has been changed in the manuscript.
-The project was approved in 2015, in 2020 is finish or they are recruiting patients?
Patients are currently being recruited and the preliminary results are very promising, so we hope that we will soon have all the data to be able to publish and disseminate results.
-Please define in material and methods, the Harvey-Bradshaw index
Table 1 provides a short explanation of the Harvey-Bradshaw Index.
-What is the dietary record, 24h, or FFQ?
We used a 24-h dietary record and the following information is included in the Study design section:
“The patients will be provided with a 24-h dietary survey designed to record their dietary intake 3 days prior to stool collection at inclusion, which will be repeated at 6 months to minimize bias concerning diet. Similarly, the patients must complete a validated survey of adherence to the Mediterranean diet at inclusion and at 6 months to assess significant changes in their eating habits”.

Round 2
Reviewer 2 Report
Authors addressed my concerns properly. No comments further.
Author Response
Dear Reviewer 2,
We do not see any new comments from you.
Do not hesitate to contact us if you have any questions. We look forward to hearing from you soon.
Best regards,
Juan Martinez Blanch
Reviewer 3 Report
Thank you to the authors for taking into account my previous comments. In general, the manuscript has improved and I encourage to the authors use the template to the journal. All my comments were properly addressed
Best regards
Author Response
Dear Reviewer 3,
We do not see any new comments from you.
Do not hesitate to contact us if you have any questions. We look forward to hearing from you soon.
Best regards,
Juan Martinez Blanch